# Risk Factors of Proximal Junctional Kyphosis in Adolescent Idiopathic Scoliosis—The Spinous Processes and Proximal Rod Contouring

**DOI:** 10.3390/jcm11206098

**Published:** 2022-10-16

**Authors:** Kathrin Boeckenfoerde, Albert Schulze Boevingloh, Georg Gosheger, Sebastian Bockholt, Lukas Peter Lampe, Tobias Lange

**Affiliations:** 1Department of Orthopedics and Tumor Orthopedics, Muenster University Hospital, 48149 Muenster, Germany; 2Department of Orthopedics and Trauma Surgery, St. Josef-Hospital, Ruhr-University Bochum, 44791 Bochum, Germany

**Keywords:** adolescent idiopathic scoliosis, proximal junctional kyphosis, complication, risk factor, spinous process, sagittal alignment, rod contouring

## Abstract

Proximal junctional kyphosis (PJK), as one of the most discussed problems after corrective surgery in adolescent idiopathic scoliosis (AIS), is still not fully clarified and seems to be multifactorial. Biomechanical and a few clinical studies have shown the influence of destruction of posterior ligaments by resection of spinous processes and some parameters concerning rod contouring as risk factors for PJK. To verify these results, 192 patients with AIS and corrective surgery via a posterior approach between 2009 and 2017 were included. Radiographic parameters were analyzed preoperatively (preOP), postoperatively (postOP), and with a mean follow up (FU) of 27 months. The participants were divided into two groups (PJK group and non-PJK group). The incidence of PJK was 15.6%. Contrary to the results of biomechanical studies, we could not find any significant influence of the spinous process resection. However, the PJK group had significantly larger preOP T4–T12 kyphosis (31.1° ± 13.93° vs. 23.3° ± 14.93°, *p* = 0.016). Furthermore, the PJK group showed a significantly larger rod contour angle (RCA) (8.0° ± 4.44° vs. 5.9° ± 3.28°, *p* = 0.003) and mismatch of postOP proximal junctional angle (PJA) and RCA (3.5° ± 5.72° vs. 0.9° ± 4.86°, *p* = 0.010) compared to the non-PJK group. An increase in the mismatch of postOP PJA and RCA (OR = 1.14, *p* = 0.008) and a high RCA are risk factors for PJK and need to be focused on by surgeons.

## 1. Introduction

Adolescent idiopathic scoliosis (AIS) is a three-dimensional spinal deformity primarily handled by conservative treatment with bracing and physiotherapy. In case of severe progression of the main curve >40–50° Cobb angle, surgical management is a generally agreed recommended option [1]. While different surgical approaches exist, posterior correction and instrumentation of the spine is one of the most frequently performed surgical procedures [1]. Reducing long-term sequelae arising from spinal deformity surgery is targeted in spinal research by identifying risk factors of complications. However, complications arise in 1.5 to 5.7% of cases of corrective surgery of AIS [2,3]. One of the currently most discussed problems is proximal junctional kyphosis (PJK), a radiological finding that can manifest itself in the clinical problem of proximal junctional failure. Depending on the used definition of PJK, the incidence varies between 14 and 46% [4,5,6,7,8,9,10,11] with a revision rate of about 10% [12]. The etiology of PJK seems to be multifactorial and has not yet been conclusively clarified. Numerous aspects, including the surgical technique, different types of implants, and stiffness of rod material, as well as preoperative (preOP) thoracic hyperkyphosis, further parameters related to the sagittal profile or certain types of AIS according to the Lenke classification, have been discussed to date as risk factors in the literature [6,13,14]. In addition, it was recently shown that two interesting parameters representing the spine’s sagittal alignment at the proximal part of the instrumentation, namely the rod contouring angle (RCA) and its mismatch with the proximal junctional angle (PJA), are associated with the occurrence of PJK [9,15]. Decreased RCA, representing the amount of kyphotic curvature bent into the rod, has been suggested to be a risk factor for PJK, but the importance of a mismatch between PJA and RCA (PJA-RCA) has not been fully understood and has rarely been studied in AIS [15]. Furthermore, spinous processes are often sacrificed during posterior surgery harvesting cancellous bony material for spondylodesis. The simultaneous resection of the posterior ligaments could lead to oversized flexibility and thus facilitate PJK. Therefore, the integrity of posterior ligaments has been recently discussed as a potential risk factor in biomechanical studies [16].

This single-center study aims to identify potential risk factors of PJK in AIS, particularly spinous process resection, and whether careful contouring of the proximal rod representing the kyphotic curvature of the spine is important in preventing PJK.

## 2. Materials and Methods

This single-center study is a retrospective analysis of 192 patients with corrective surgery via a posterior approach using the same screw rod system (titanium screws, titanium and/or CoCr rods, Expedium^®^, DePuy Synthes) and the same surgical technique in all patients for AIS between 2009 and 2017. The study protocol was approved by the local ethics commission. As this study was a retrospective study of pre-existing radiologic (hospital PACS system) and clinical data (patient record), there was no need to educate patients about the conduct of the study.

Full spine X-ray films were analyzed preOP, postOP, and at a mean follow up (FU) of 27 months (m) in anteroposterior and lateral standing for all cases. All X-rays were digitally stitched. The radiographic parameters were measured by a single observer (T.L.), all other parameters were retrieved by medical record review.

Clinical data included age at surgery, weight, height, gender, and Risser sign. Radiographic parameters included the following spinal parameters: pre- and postOP Cobb angles, the T4–T12 kyphosis (TK), and the L1–S1 lordosis (LL). Additionally, the level of upper (UIV) and lower (LIV) instrumented vertebrae, number of instrumented vertebrae, the RCA (angle between UIV and UIV-1) postOP and the mismatch between RCA and the PJA, the angle between the caudal endplate of UIV and the cranial endplate of UIV + 2 were assessed. Furthermore, sagittal parameters such as pelvic tilt (PT), sacral slope (SL), pelvic incidence (PI), and sagittal vertical axis (SVA) were analyzed.

PJK is defined as a kyphosis characterized by two requirements: (1) segmental kyphosis between UIV and UIV + 2 ≥ 10° and (2) the postoperative segmental kyphosis between UIV and UIV + 2 is at least 10° greater than preoperative [15] (Figure 1).

Based on this definition, patients were divided into two groups (PJK group with *n* = 30 and non-PJK group with *n* = 139).

Statistical analyses were performed in IBM SPSS Statistics, version 26.0 (IBM Corporation, Armonk, New York, NY, USA). First, the incidence of PJK was assessed using cross tables. Differences between the two groups (PJK group/non-PJK group) were investigated using a Student *t*-test, and Mann–Whitney-U test for continuous data. The chi-square test was used for categorial data. We also assessed the Pearson correlation between all significant predictors. If one of these tests indicated that the independent variable significantly influences the incidence of PJK, it was included in the subsequent multivariate logistic regression analysis. If parameters were highly correlated (>|0.7|) among each other, only the one with the highest correlation to the dependent variable was included in the model. Using this analysis approach, we aimed at predicting the probability of PJK occurrence based on the value of continuous and categorical variables. Descriptive values are described as mean ± standard deviation for metric data. The alpha level was assigned as *p* < 0.05.

## 3. Results

A total of 192 AIS patients with a mean age of 14.7 ± 6.25 years were included in this study of which 77.1% (*n* = 148) were female. Twenty-three participants were not analyzed due to loss to FU. Thirty of 169 patients (15.6%) developed PJK (PJK group) while 139 (84.4%) showed no PJK (non-PJK group) with a mean FU of 27 m. Eighty percent (*n* = 24) of all patients developed PJK within the first 12 m, whereas only 20% (*n* = 6) developed PJK after a longer FU period of more than 24 m. The UIV was distributed as follows: T3 (12.5%, *n* = 23), T4 (20.8%, *n* = 40), T5 (22.4%, *n* = 43), T6 (9.4%, *n* = 18). The LIV was distributed as follows: T12 (16.1%, *n* = 31), L1 (14.1%, *n* = 27), L2 (10.9%, *n* = 21), L3 (27.6%, *n* = 53), L4 (24.5%, *n* = 47). An average of ten vertebrae were instrumented in each patient.

Age, sex, body mass index, number of instrumented vertebrae, and preOP and postOP major Cobb angle were similar in both groups (Table 1).

While 35.4% (*n* = 68) of patients received bilateral titanium rods, 48.4% (*n* = 93) bilateral cobalt–chrome rods, and 16.1% (*n* = 31) a combination of both materials, there is no significant difference in PJK frequency regarding rod material in both groups.

Spinous process resection and, if so, the number of resected processes does not show any increase in the rate of PJK (Table 1).

Patients with PJK had a significantly higher preOP TK (31.1° ± 13.93° vs. 23.3° ± 14.93°, *p* = 0.016). Furthermore, the non-PJK group showed a tendency to a slightly better TK preservation (+ 0.4° vs. −1.5°, *p* = 0.527) in contrast to the PJK group with a slight flattening of the TK. However, postOP TK was still significantly higher in the PJK group than in the non-PJK group. Moreover, spino-pelvic alignment parameters (SVA, PI, PT, SL) showed no difference between the two groups (Table 1).

Immediately postOP, PJA and RCA significantly differed between both groups (PJA: 11.6° ± 5.37° (PJK group) vs. 6.8° ± 4.53 (non-PJK group), *p* < 0.001; RCA: 8.03° ± 4.44° vs. 5.9° ± 3.28°, respectively *p* = 0.003). The mismatch between PJA and RCA (PJA-RCA) was significantly higher in the PJK group (3.5 ± 5.72 vs. 0.9 ± 4.86, *p* = 0.010), representing a risk factor.

Before running the multiple logistic regression analysis, correlations amongst predictors were explored (Table 2).

All predictors were included in the analysis because none of the predictors were highly correlated (>|0.7|). In the multiple logistic regression analysis, predicting the risk of PJK was based on TK, RCA, and the mismatch PJA-RCA. The RCA impacted the risk for PJK at a mean FU of 27 m (OR = 1.26, *p* < 0.001), suggesting that a one degree increase in RCA yields a 1.3 times higher likelihood of PJK and, furthermore, a one degree increase in the mismatch PJA-RCA (OR = 1.14, *p* = 0.008) accounted for a 14% increased risk of developing PJK (Table 3).

## 4. Discussion

PJK, especially the more advanced form, PJF, is a complication after corrective surgery with posterior instrumentation of the spine in AIS patients with either the potential risk of need for revision surgery or health quality burden. Depending on the definition employed in previous studies, the incidence of PJK varies between 14% and 46% [4,5,6,7,8,9,10,11]. The big variation in the literature comes from having studies analyzing different types of scoliosis, such as AIS, neuromuscular, or adult spinal deformities. Different and more risk factors have been identified in adults and patients with neuromuscular scoliosis than in patients with idiopathic scoliosis. Furthermore, different definitions of PJK exist. Helgeson et al., measuring PJA at UIV and UIV + 1, reported a lower number of PJK incidents (8.1%) and Rhee et al. used the more common definition that we used, measuring between UIV and UIV + 2, and reported a higher number of PJK cases (35%) [7,10]. In the present study, 15.6% of all patients developed PJK. Only AIS patients were included, which could be partly responsible for the lower PJK incidence.

Previous studies evaluated a variety of risk factors that might be associated with PJK. Destructed posterior ligaments, poor bone quality, and sagittal imbalance have been shown to increase the risk for PJK [17,18,19]. Others revealed that posterior and anterior corrective instrumentation, excessive reduction of TK, a high absolute thoracic kyphosis, thoracoplasty, as well as several Lenke subtypes may be associated with PJK [6,7,13].

In a biomechanical study, Lange et al. suggest that destabilization of posterior ligaments at the adjacent segment to an instrumentation and even at the level of instrumentation might be related to a higher incidence of PJK due to the fact that supraspinous ligaments are multisegmental [20]. Cammarata et al. also describe a positive correlation of a dissection of posterior ligaments with the increase in the stress level, which eventually leads to PJK in their computer simulation study [17]. In contrast, the results of this clinical study showed no difference in the amount or level of resected spinous processes with the posterior ligaments within the instrumented levels in the PJK and non-PJK group. However, it is common and important to preserve the supraspinous and interspinous ligament complex at the adjacent UIV/UIV + 1 level which we obviously did in our patients [21,22]. However, Arlet and Aebi postulated for adult spinal deformities a potential elongation through stretching of the posterior ligament complex or even a disruption of the ligaments eventually combined with an insufficiency fracture of the spinous process. These risk factors in the postoperative course can hardly be influenced by the surgeon [23].

Highlighting the importance of the sagittal rod contouring, the mismatch between PJA and RCA as a risk factor for PJK is evaluated. Lonner et al. presume the RCA, chosen and applied by the surgeon, influences the segment proximal to the spinal instrumentation [13]. Our study confirms further evidence from a biomechanical study of Cammarata et al. who demonstrate that an increase in the RCA from 10° to 20°, 30°, and 40° increased the PJA by 6%, 13%, and 19%, suggesting that an inappropriate sagittal rod curvature with an overbending yields PJK [17]. Our results show that a one degree increase in RCA accounts for a 1.3 times increased risk for developing PJK and this means that a general overbending of the rod at the proximal end should be avoided. This overbending is reflected by the PJA which in our study is much higher in the PJK group immediately postOP than in the non-PJK group, meaning that the surgeon in some cases obviously overbent the rod compared to the non-PJK group, while there is no PJA difference in both groups preOP. On the other hand, a too straight rod at the proximal end also seems to be a risk factor, as indicated by a mismatch between PJA and RCA. Our logistic regression analysis suggests that a one degree increase in the mismatch is associated with a 14% increased likelihood to suffer from PJK. It should not be disregarded that RCA might be influenced by different variables such as rod material, strength of the rod, and the screw angle below. Still, our results are in line with Wang et al., who describe the mismatch as a relative straightness of the rod compared to the thoracic kyphosis. They highlight the importance of proper rod contouring to prevent a PJA-RCA mismatch [9].

Surgeons should therefore pay attention to the given alignment and be aware of a higher risk of PJK by modifying the patient’s alignment to be too straight or too kyphotic.

Before surgery, patients with scoliosis subconsciously compensate their three-dimensional disbalance with a minimum of energy. This can be attributed to the concept of the “cone of economy” described by Dubousset, which refers to a person’s stable region of upright standing posture [24]. During surgery, the patient’s sagittal balance may be disturbed, thereby necessitating new compensation strategies. One of these strategies may be enhanced muscle tonus or hip and knee bending. If such compensation strategies cannot be maintained, PJK may develop [25]. To better understand the influence of sagittal balance on the development of PJK, we first examined the relation between TK and the incidence of PJK. Second, we included SVA as an often-mentioned risk factor for PJK. Our results showed that the preOP TK was significantly higher in the PJK group than in the non-PJK group. In accordance with Kim et al., our logistic regression analysis confirmed that a one degree increase in TK corresponded to a 10% increase in the risk of PJK [6]. This finding is in line with Lonner et al. [13]. Therefore, the operative surgeon should mind the higher complication risk. In contrast to Kim et al., we did not find any significant evidence that a strong surgical correction of TK would promote PJK [6]. As mentioned before, we also included SVA to evaluate the sagittal alignment. In contrast to Wang et al., but in line with Burton et al. [26], our data do not indicate a significant difference in the SVA scores between the PJK group and the non-PJK group [9]. This can be explained by the fact that in our study there is no difference in the SVA preOP in both groups at all whereas Wang et al. found a difference in the preOP SVA in their PJK and non-PJK groups. Basically, one should assume that otherwise healthy patients with idiopathic scoliosis do not suffer from abnormal decompensated preOP SVA as we could show in our cohort.

Our analysis includes all types of Lenke classification, grouped along the line of Lonner et al. [13]. We do find differences in PJK incidence between the individual Lenke types, but these are not statistically significant. However, Lonner et al. demonstrate that risk factors for PJK differ between the different Lenke types [13]. Due to the larger sample size in their study, with more data for each Lenke group, they may have found significant differences which we could not find in our study.

There are several limitations in this study. First, the retrospective design limits the analysis. Second, despite standardization, there is a certain variability in body position when taking X-ray images, so a bias, albeit small, must be expected here. One possible solution for future studies would be to use a whole-body X-ray system, such as EOS, so that image acquisition is fast and continuous and, at the very least, multiple images do not have to be acquired and then digitally stitched together. Thirdly, this study only includes radiographic parameters to characterize PJK. There are no clinical data such as pain and mechanical instability to allow distinction between PJK and PJF. Obviously, the PJA-RCA difference only assesses the risk for PJK as a radiological diagnosis. The clinical importance and relevance regarding the need for surgical revision can be calculated with the Hart ISSG severity scale.

Nevertheless, compared to the literature, this clinical study has a large sample size arising from only a single center, limiting biases due to possible different surgical techniques and procedures in different centers, and focuses only on AIS patients. In addition, only current and complete patient data are included in this analysis.

## 5. Conclusions

In conclusion, this study reveals several risk factors for PJK, such as a high preOP thoracic kyphosis, a too high RCA and especially a high mismatch of PJA and RCA, representing an improper rod contouring compared to the natural and individual kyphosis of the spine. In contrast, resection of spinous processes is no risk factor for PJK.

Consequently, surgeons need to pay attention to proper rod contouring, especially in the proximal part of instrumentation, to maintain the individual sagittal spinal alignment. The rod should be neither too straight nor overbent but should optimally match the PJA. This is especially true for patients with preoperatively pronounced thoracic kyphosis.

## Figures and Tables

**Figure 1 jcm-11-06098-f001:**
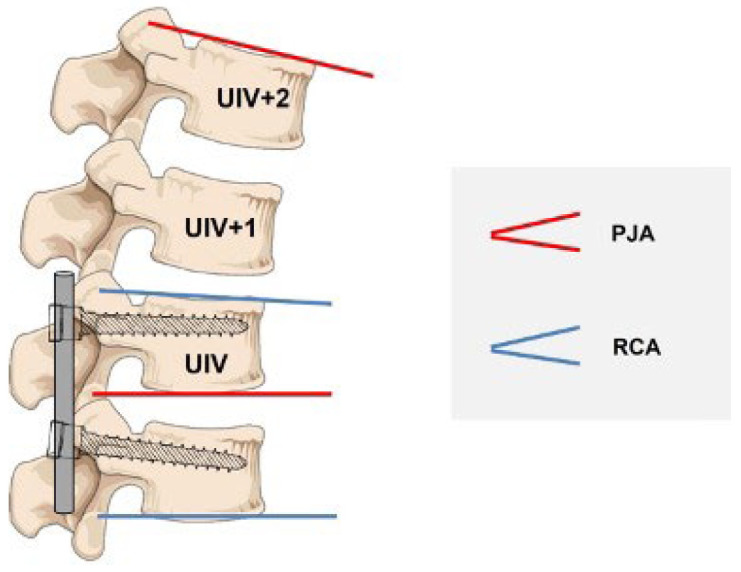
The measurement of proximal junctional kyphosis (PJK) and rod contour angle (RCA).

**Table 1 jcm-11-06098-t001:** Demographic, radiographic, and surgical data (data shown as mean ± standard deviation (SD)).

	Non-PJK Group	PJK Group	*p*
Age	16.1 ± 4.36	16.9 ± 8.66	0.416
BMI	20.9 ± 3.95	20.6 ± 3.20	0.747
UIV (median)	T4	T5	0.248
Instrumented vertebra (*n*)	10.4 ± 2.55	10.5 ± 2.52	0.901
Percentage of patients with resected spinous processes	71.5%	73.3%	0.843
Resected spinous processes (*n*)	5.6 ± 4.15	6.1 ± 4.42	0.521
Cobb preOP (°)	62.3 ± 15.06	61.9 ± 11.63	0.889
Δ Cobb preOP vs. postOP (°)	38.2 ± 9.92	37.5 ± 10.59	0.728
TK preOP (°)	23.3 ± 14.93	31.1 ± 13.93	0.016
TK postOP (°)	23.7 ± 10.01	29.6 ± 9.73	0.004
TK 12 m FU (°)	25.5 ± 10.64	35.2 ± 11.2	<0.001
Δ TK preOP v. postOP (°)	0.4 ± 11.7	−1.5 ± 8.12	0.527
LL preOP (°)	45.0 ± 11.54	45.7 ± 13.35	0.767
LL postOP (°)	41.0 ± 11.48	44.4 ± 13.83	0.155
LL 12 m FU (°)	45.75 ± 12.18	46.83 ± 13.8	0.668
PI (°)	52.6 ± 12.81	51.7 ± 11.26	0.727
PT (°)	10.3 ± 7.1	9.9 ± 6.97	0.773
SL (°)	42.7 ± 10.51	41.8 ± 9.75	0.679
SVA preOP (cm)	0.8 ± 2.88	0.8 ± 2.52	0.938
SVA postOP (cm)	1.8 ± 2.84	2.9 ± 2.83	0.066
SVA 12 m FU (cm)	0.9 ± 2.73	1.3 ± 2.85	0.409
Δ SVA (cm)	−1.0 ± 3.24	−2.0 ± 2.90	0.113
PJA preOP (°)	5.4 ± 4.88	6.5 ± 5.50	0.387
PJA postOP (°)	6.8 ± 4.53	11.6 ± 5.37	<0.001
PJA 12 m FU (°)	7.6 ± 4,76	19.37 ± 8.11	<0.001
RCA (°)	5.9 ± 3.28	8.0 ± 4.44	0.003
postOP PJA-RCA (°)	0.9 ± 4.86	3.5 ± 5.72	0.010

Note: UIV, upper instrumented vertebra; TK, thoracic kyphosis (T4–12); LL, lumbar lordosis (L1–S1); PI, pelvic incidence; PT, pelvic tilt; SL, sacral slope; SVA, sagittal vertical axis; PJA, proximal junctional angle; RCA, rod contour angle; postOP PJA-RCA, mismatch between postOP PJA and RCA. Examining the individual Lenke groups, there was no significant difference in PJK frequency.

**Table 2 jcm-11-06098-t002:** Pearson correlation analysis.

	1.	2.	3.	4.	5.	6.
1. postOP PJA-RCA	1					
2. preOP PJA-RCA	0.634 **	1				
3. RCA	0.360 **	0.472 **	1			
4. TK 12 m FU	−0.255 **	0.002	0.192 *	1		
5. ΔTK	0.204 **	0.324 **	0.137	−0.065	1	
6. PJK 12 m FU	−0.423 **	−0.206 **	0.300 **	0.424 **	−0.204 **	1

* *p* < 0.01, ** *p* < 0.001. Note: postOP PJA-RCA, mismatch between postOP PJA and rod contour angle; preOP PJA-RCA, mismatch between preOP PJA and RCA; RCA, rod contour angle; TK, thoracic kyphosis T4–T12; ΔTK, thoracic kyphosis preOP–thoracic kyphosis postOP.

**Table 3 jcm-11-06098-t003:** Risk factors for PJK in multivariate analysis.

	Estimate Coefficient	Odds Ratio (Exp(B))	*p*	95% Confidence Interval Lower Upper	
postOP PJA-RCA	0.134	1.143	0.008	1.036	1.262
RCA	0.228	1.257	0.001	1.095	1.442
TK preOP	−0.008	0.992	0.730	0.950	1.037
TK postOP	−0.039	0.962	0.320	0.891	1.038
TK 12 m FU	0.095	1.100	0.011	1.022	1.184

Note: postOP PJA-RCA, mismatch between postOP PJA and RCA; RCA, rod contour angle; TK, thoracic kyphosis T4–T12.

## Data Availability

Data collected for this study, including individual patient data, will not be made available.

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
