# Peer review of "Risk Factors of Proximal Junctional Kyphosis in Adolescent Idiopathic Scoliosis—The Spinous Processes and Proximal Rod Contouring"

_jcm, 2022, doi:10.3390/jcm11206098_

Round 1
Reviewer 1 Report
The major flaw in the work is the lack of complete references, the failure to count the effect size and the sample size. Please also refer to the other specific comments that I have listed below.
1. Editing error: There should be a quote at the end of the sentence and then a period. Note applies to all text. For example:
2. L32 Change ‘’ recommended option.[1]” on ‘’recommended option [1]. ‘’
3. L57 –‘’ large numbe’’ - it is very subjective. I suggest removing these words.
4. L65 - Were patients aware that they were participating in the study and that they could opt out at any time? Please indicate this in the text.
5. L68 – ‘’ single observer’’ - If it is one of the authors, please indicate him. Was it a specialist in radiology ?
6. L85 - Statistical analyses - Please add the effect size and add sample size calculations.
7. In the version of the manuscript that is available in the mpdi system, there are 17 items of bibography. The authors in the main text have references to 27 (L218 ). It is necessary to fill in the gaps.
I declare no evaluation in terms of discussion due to incomplete bibography.
I declare no evaluation in terms of self-citation due to incomplete bibography.
Thank you for the opportunity to evaluate.
Reviewer 2 Report
In this article, the authors explored risk factors of proximal junctional kyphosis (PJK) in adolescent idiopathic scoliosis. They especially focused on proximal rod contouring, and revealed that postoperative rod contour angle (RCA) and proximal junctional angle (PJA)-RCA mismatch were risk factors for PJK. I think the article is well-written; however, there are some concerns which have to be solved.
Major criticism
1. (Overall) In this clinical study, authors reported that high preoperative thoracic kyphosis, high RCA, and PJA-RCA mismatch were risk factors for PKJ. They have been already reported as risk factors in the previous reports; therefore, the present study has less novelty as it is.
2. (Title and Introduction) Authors titled this article “Risk Factors of Proximal Junctional Kyphosis in Adolescent Idiopathic Scoliosis – the Spinous Processes and Proximal Rod Contouring”, which may cause someone to believe that spinous process resection is a risk factor of PJK. And they described that the purpose of the study was “to identify potential risk factors of PJK in AIS, particularly spinous process resection and…” (P2, L57-58). However, in this study, spinous process resection was not a risk factor of PJK in AIS. I think the current title is somewhat misleading. In addition, now that spinous process resection was focused on in the aim of study, authors should refer to spinous process resection in conclusion.
3. (Results) Authors showed that “the number of resected spinous processes does not show any increase in the rate of PJK” (P4, L119-120). I guess that not only the number of resected spinous processes but also the levels of resected spinous processes are important to discuss the risk factor of PJK. It may be valuable to compare two groups whether the upper spinous process was resected or not.
4. (Results) Authors showed that high RCA and PJA-RCA mismatch were risk factors for PKJ. However, it is quite complicated because RCA and PJA-RCA are influence each other. If authors can suggest the specific range of RCA which surgeons should target, it will be valuable.
Minor criticism
1. (Materials and Methods) Authors described the corrective surgeries were performed “using the same screw rod system” (P2, L63), but some different materials were used (P4, L116-117). Please explain this contradiction.
2. (Materials and Methods) In this study, the radiographic analyses were performed before operation, immediately after operation, and at follow up period whose mean was 27 months after operation. How much different were follow up periods among patients? As I worry that the longer follow up period was, the higher the incidence of PJK was. I’d recommend that a variation of follow up period should be added. In addition, if so, authors should show that there is no correlation between the incidence of PJK and follow up period.
3. (Result) Significant figures were not unified.
4. (Result) There are some typos in Table 2 (“0-.065 and 0-.204”).
5. (Result) 95% confidence interval should be added in Table 3.
(References) Ref. 18-27 are not found.
Round 2
Reviewer 1 Report
Thank you for sending the corrected version. The author's corrections and explanations are acceptable. I have one comment that he suggests before publishing (L62-63) - please provide the exact specifications of the screws.
Reviewer 2 Report
The manuscript have been revised well.
With regard to the concern about novelty, I understood authors' argument.
I could also understand that some patients were not resected spinous processes at all, whereas, the others were resected ones at all levels of instrumentation. Then, I think that two groups, non-PJK and PJK group, should be compared by the percentage of patient who were resected spinous processes, rather than the number of resected resected spinous processes.
